# DEAD Box Helicase 24 Is Increased in the Brain in Alzheimer’s Disease and *App^N-LF^* Mice and Influences Presymptomatic Pathology

**DOI:** 10.3390/ijms25073622

**Published:** 2024-03-23

**Authors:** Michael Axenhus, Tosca Doeswijk, Per Nilsson, Anna Matton, Bengt Winblad, Lars Tjernberg, Sophia Schedin-Weiss

**Affiliations:** 1Department of Neurobiology, Care Sciences and Society, Division of Neurogeriatrics, Karolinska Institutet, 171 64 Solna, Sweden; michael.axenhus.2@ki.se (M.A.); tosca.doeswijk@ki.se (T.D.); per.et.nilsson@ki.se (P.N.); anna.matton@ki.se (A.M.); lars.tjernberg@ki.se (L.T.); 2Faculty of Psychology and Neuroscience, Maastricht University, 6211 LK Maastricht, The Netherlands; 3Campus Huddinge, Theme Inflammation and Aging, Karolinska University Hospital, 141 57 Huddinge, Sweden

**Keywords:** Alzheimer’s disease, DDX24, hippocampus, immunohistochemistry, post-mortem brain

## Abstract

At the time of diagnosis, Alzheimer’s disease (AD) patients already suffer from significant neuronal loss. The identification of proteins that influence disease progression before the onset of symptoms is thus an essential part of the development of new effective drugs and biomarkers. Here, we used an unbiased ^18^O labelling proteomics approach to identify proteins showing altered levels in the AD brain. We studied the relationship between the protein with the highest increase in hippocampus, DEAD box Helicase 24 (DDX24), and AD pathology. We visualised DDX24 in the human brain and in a mouse model for Aβ42-induced AD pathology—*App^NL-F^*—and studied the interaction between Aβ and DDX24 in primary neurons. Immunohistochemistry in the AD brain confirmed the increased levels and indicated an altered subcellular distribution of DDX24. Immunohistochemical studies in *App^NL-F^* mice showed that the increase of DDX24 starts before amyloid pathology or memory impairment is observed. Immunocytochemistry in *App^NL-F^* primary hippocampal neurons showed increased DDX24 intensity in the soma, nucleus and nucleolus. Furthermore, siRNA targeting of DDX24 in neurons decreased APP and Aβ42 levels, and the addition of Aβ42 to the medium reduced DDX24. In conclusion, we have identified DDX24 as a protein with a potential role in Aβ-induced AD pathology.

## 1. Introduction

Plaques formed by the amyloid β-peptide (Aβ) and neurofibrillary tangles (NFTs) composed of hyperphosphorylated tau protein are associated with cognitive decline and brain pathology in Alzheimer’s disease (AD) [1,2,3] and are hallmarks of the disease [4]. Aβ is formed by proteolytic processing of the amyloid precursor protein (APP) [5,6]. The Aβ variant containing 42 amino acids (Aβ42) is aggregation-prone and important in the development of AD [7,8,9]. The current consensus within the AD field is that a reduction of Aβ pathology might help prevent cognitive decline. In support of this notion, targeting Aβ pathology directly via antibodies has yielded two drugs approved for the treatment of AD: Aducanumab and Lecanemab [10]. There are also several immunotherapies in phase 3 trials.

Although aggregation of Aβ is an early event in AD, the disease is multifactorial and influenced by genetics and environmental factors, suggesting that other pathways may also be involved in AD pathology [11]. Moreover, at the time of diagnosis, AD-associated Aβ pathology is already advanced with a large amyloid load in the brain [12]. Thus, drugs need to be designed to target the presymptomatic stage of the disease. By identifying dysregulated proteins and pathways within the AD brain, novel strategies for pharmaceutical intervention may be discovered [13].

We used proteomics and Ingenuity Pathway Analysis (IPA) to find proteins with altered expression in postmortem hippocampus and frontal cortex from AD cases in a similar manner as described in our recent study using the *App^NL-F^* AD mouse model [14]. The protein with the highest alteration in the hippocampus in AD brain was the ATP-dependent RNA helicase DDX24 (DDX24), a member of the DEAD-box protein family. This family consists of a large group of putative RNA helicases that mediate nucleoside triphosphate-dependent unwinding of double-stranded RNA. These proteins are implicated in cellular processes involving alteration of RNA secondary structure, such as translation initiation, RNA splicing, ribosome assembly and spliceosome assembly [15]. Studies based on distribution patterns suggest that these proteins are involved in embryogenesis, cellular growth and division [16,17]. DDX24 knock-down mice do not survive past the embryogenic stage, suggesting that the protein plays a vital function in embryonic development [18]. It is also implicated in the innate immune response, where DDX24 has been found to regulate lymphocytes, and higher levels of DDX24 are typically indicative of an increased pro-inflammatory response [19,20]. From a disease perspective, the protein has mostly been studied in association with oncological disorders [21]. 

We used immunohistochemistry (IHC) and confocal microscopy to confirm that the levels of DDX24 are altered in AD brain, and we followed the progression of DDX24 levels in the brain tissue of *App^NL-F^* mice. This study shows that siRNA silencing of DDX24 in mouse primary neurons was accompanied by decreased amounts of APP and Aβ and a reduced number of synapses. Moreover, we found that Aβ concentrations decreased in conjunction with the DDX24 signal, while high Aβ exposure to the neurons induced depression of the DDX24 signal. We suggest that DDX24 is involved in AD pathogenesis at an early stage of the disease. 

## 2. Results

### 2.1. Proteomics and IPA Analysis

The relative abundance of individual proteins in the AD brain compared to the non-demented (ND) brain was analysed by using ^18^O-labeling followed by nano-HPLC MS/MS. Around 3000 proteins were quantified in each sample. Among those, 266 and 223 proteins were significantly altered (*p* ≤ 0.05) in the hippocampus and cortex, respectively, in AD compared to the ND brain. The ten proteins with the largest fold change (increase or decrease) in AD compared to ND brain are shown for the hippocampus and cortex (Table 1). Among these, DDX24 had the highest protein increase in the AD hippocampus compared to ND (Table 1). All proteins with at least a 1.1-fold change and *p* < 0.05 were analysed by IPA. In both the hippocampus and cortex, the synaptogenesis signalling pathway was among the top pathways that differed between AD and ND. Interestingly, DDX24 was a prominent protein in the highest-ranked network (denoted cellular assembly and organisation, cancer, developmental disorder) (Figure 1).

### 2.2. Immunohistochemistry Showed Increased Levels of DDX24 in AD Brain

Having identified DDX24 as a potential player in AD pathology, we set out to examine in detail its expression in the brain of AD patients. Firstly, we stained thin sections from AD and ND brains for DDX24 and analysed the resulting signal. Sections derived from AD brains were characterised by an advanced stage of neurodegeneration (Braak stage V or VI), whereas ND brains showed little pathological hallmarks (Braak stage 0-II). We quantified DDX24 within neurons by measuring the signal intensity in individual cells (*n* = 40) in each of the three experiments (*n* = 3). Hippocampal pyramidal neurons within both CA1 and CA3 showed a marked elevation of DDX24 in the AD brain (Figure 2A,B). No significant DDX24 signal was found in astrocytes or microglia. ND samples showed weak cytosolic staining and intense nuclear staining in structures around 3–6 µm, suggesting location in the nucleolus (Figure 2C,D). In the AD brain, the cytosolic staining was more intense in many neurons. These neurons with more intense staining appeared to be more dystrophic than other neurons (Figure 2B–D). The nucleolar staining was significantly higher in the AD brain (Figure 2C,D). We also observed staining in neurites, though no significant difference was found. In our immunocytochemistry experiments described later, we also saw staining in the neurites, suggesting the background staining in the tissue largely consists of neuritic staining.

### 2.3. Confocal Microscopy Showed no Colocalisation of DDX24 with Pathologic Tau 

Given that DDX24 was noted intracellularly within hippocampal pyramidal neurons, we investigated whether intracellular tau tangles colocalised with DDX24. We, therefore, examined the colocalisation of DDX24 and hyperphosphorylated tau using the well-characterised AT8 antibody, which binds to phosphorylated regions in the tau protein at serine 202, threonine 205 and serine 208. No colocalisation of DDX24 and phosphorylated tau was observed in pyramidal neurons (Figure 3). 

### 2.4. Studies in an AD Mouse Model Showed a Presymptomatic Increase in DDX24 

To gain insight into the accumulation of DDX24 during the progression of Aβ pathology in a mouse model of Aβ pathology, we stained 6-, 12-, and 25-month-old *App^NL-F^* and WT mice brains for DDX24. Using IHC, we found that DDX24 was detectable throughout the hippocampus and in the cortex of WT and *App^NL-F^* mice (Figure 4 and Appendix A). DDX24 was significantly increased in *App^NL-F^* at 6 months of age, i.e., before the appearance of Aβ plaques and the onset of cognitive impairment. The increased DDX24 levels in 6-month-old *App^NL-F^* mice were found in subareas rich in pyramidal neurons such as the CA1 and CA3 (Figure 4A,B), in the molecular layer of the dentate gyrus (Figure 4C) and in the cortex (Figure 4D). There was no significant change in DDX24 staining intensity in the brain of *App^NL-F^* mice aged 12- or 25 months old when compared to WT, although there was a tendency towards an increase in DDX24 in 12-month-old *App^NL-F^* mice (Appendix A). While the DDX24 staining intensity increased with age in WT mice in all brain regions investigated, it changed less over time in the *App^NL-F^* mice (Figure 5). Thus, the increased levels of DDX24, observed in presymptomatic *App^NL-F^* mice at 6 months of age compared to WT mice, became less prominent over time and were not present in aged mice with cognitive defects. DDX24 in mouse brains appeared to be located mostly in the soma and the nucleolus of neuronal cells, whereas the staining intensity was less pronounced in neurites (Appendix A). 

### 2.5. DDX24 Accumulated in the Soma, Nucleus and Nucleolus of Primary Neuronal Cultures Derived from App^NL-F^ Mice

Identifying the localisation of the DDX24 accumulation in *App^NL-F^* mice could clarify the metabolism and trafficking of the protein. To this end, we studied the presence of DDX24 in primary neuronal cultures derived from *App^NL-F^* and WT mice cultured for 7, 14 and 21 days in vitro (DIV). We measured the signal intensity in the soma, nucleus, nucleolus and neurites of the cells. DDX24 was increased in the soma and nucleus of *App^NL-F^* hippocampal neurons compared to WT at 7 and 14 DIV but not at 21 DIV (Figure 6A–F). DDX24 was also increased in the nucleolus at 7, 14 and 21 DIV in *App^NL-F^* neurons (Figure 6A–F). There was no difference in DDX24 in the neurites of *App^NL-F^* and WT neurons (Appendix A).

### 2.6. siRNA-Mediated Silencing of DDX24 in Mice Cortical Primary Neurons Reduces APP and Aβ42 Levels and Decreases the Number of Synapses

To gain insight into the effect of reduced DDX24 expression, we treated 12 DIV cortical primary cultures with siRNA targeting DDX24. To investigate whether DDX24 affects amyloid production and neurons, we measured the signal intensity of DDX24 and APP using confocal microscopy and image analysis. Neuronal status was evaluated by neuronal count and spine density, and Aβ42 levels were quantified by ELISA. Cultures that had been treated with DDX24 siRNA displayed lower levels of DDX24 and, remarkably, lower APP levels compared to controls (*n* = 3) (Figure 7A). Spine density was reduced in siRNA-treated cultures, indicating reduced synapse formation or synapse loss (Figure 7B). There was no significant reduction in the number of neurons in treated cultures, indicating that silencing of DDX24 does not affect neuronal survival (Figure 7C). The levels of Aβ42 were decreased in both cell culture medium (Figure 7D) and cell lysate (Figure 7E) of cortical neurons as a result of DDX24 silencing.

Since the main pathology of *App^NL-F^* mice is the accumulation of Aβ42, we examined the effect of Aβ42 on DDX24 levels in neurons from WT mice. Adding Aβ42 into the culture medium of 12 DIV WT primary neurons induced an increase in detectable APP and/or APP C-terminal fragments as measured by a C-terminal specific APP antibody. Higher Aβ42 concentration also lowered DDX24 significantly (Figure 8A–C).

## 3. Discussion

Using an unbiased ^18^O labelling nano-HPLC MS/MS proteomics approach to measure relative levels of proteins in the hippocampus and cortex, we found DDX24 to be the most upregulated protein in AD compared to ND brain. The high levels in the hippocampus in AD, combined with a lack of previous studies linking DDX24 to AD pathogenesis, prompted us to further investigate this protein. Though other RNA helicases and RNA binding proteins, such as DEAD-box helicase 19B, are altered in a subpopulation of AD cases [20], no previous studies have identified DDX24 as a possible regulator of AD pathology. We wanted to study this protein further to understand how RNA dysregulation may be implicated in AD pathology. Pathway analysis revealed a top canonical network, which included DDX24 and other proteins involved in cellular assembly, organisation and development, suggesting that it would be interesting to study this protein at different stages of the disease. Thus, we performed immunohistochemistry in postmortem AD brains and in *App^NL-F^* mouse brains of different ages. Moreover, we studied the relationship between DDX24 and APP metabolism in primary hippocampal neurons.

IHC of human AD brain and non-demented controls confirmed increased levels of DDX24 in AD, particularly in neuronal cells in the hippocampal subareas CA1 and CA3, where the accumulation was present mainly in the soma and in distinct nuclear structures, presumably corresponding to nucleoli. A subpopulation of cells with increased staining in the soma and no staining in the nucleoli appeared more common in the AD brains. Such altered subcellular localisation could be indicative of increased RNA processing within the cytosol or deficient trafficking to/from the nucleus. These notions are in line with the known physiologic effects of DDX24, which include translation initiation, RNA splicing, cell division and growth [21]. It could also indicate a dysfunction of RNA processing, which is in line with the finding that a subtype of AD is characterised by RNA dysregulation [20]. The defective RNA processing may, in turn, influence other processes, contributing to AD pathology.

Increased intraneuronal levels of DDX24 were confirmed by IHC to be present in CA1, CA3, dentate gyrus and cortex in mice and humans. Interestingly, the peak in increased DDX24 levels was observed at 6 months of age in mice—before Aβ plaques are detectable—and was diminished over time, showing no significant increase in *App^NL-F^* compared to WT mice at 12 and 25 months of age. One plausible explanation for this difference between the mouse model and sporadic human AD cases is that the mice have increased levels of Aβ42 from birth, while sporadic AD cases start to accumulate Aβ42 at a relatively later stage during the development of the disease [22]. Thus, the pathology observed in *App^NL-F^* mice at a relatively early stage may reflect the pathology observed in AD cases at an advanced disease stage. Since DDX24 is involved specifically in the initial phase of the immune response, this early increase could also be connected to the inflammatory environment that is associated with Aβ42 [17,18,19]. We speculate that the advanced amyloid pathology and neurodegeneration decrease DDX24 levels, either through a negative feedback mechanism or because the compensatory function of DDX24 is no longer effective.

Interestingly, the increased staining of DDX24 in soma and distinct nuclear staining, presumably corresponding to nucleoli, was similar to the accumulation of huntingtin in AD brain reported in a previous study [23]. Likewise, the finding that DDX24 levels are increased at 6 months of age—before Aβ plaques are detectable—in the *App^NL-F^* mice is in line with our previous report showing huntingtin to be accumulated in *App^NL-F^* mice with a similar accumulation pattern as DDX24 [14]. Since both huntingtin and DDX24 are nucleo-cytoplasmic proteins and similarly affected in AD brain and in *App^NL-F^* mice, these data suggest that Aβ42 exerts regulatory effects on nucleo-cytoplasmic proteins. In line with these data, previous studies suggested that transport between the cytoplasm and the nucleus is deficient in neurodegenerative diseases. Deficits in key nuclear transport proteins such as DNA methyltransferase, RNA polymerase II and RAN are suggested to be impaired in AD [17,24]. In addition, nuclear transport proteins, including RAN GTAPase-activating protein 1, importin-7 and importin-5, have been reported to be upregulated in the hippocampus of *App^NL-F^* mice before the onset of amyloidosis [14]. Since the most prominent early phenotype of the *App^NL-F^* mouse is the increased Aβ42 levels, while it lacks significant tau pathology, the increased expression of DDX24 in young asymptomatic *App^NL-F^* mice could be due to high Aβ42 levels [22].

Our study describes the cellular localisation of DDX24 in primary hippocampal neurons derived from *App^NL-F^* mice and shows that DDX24 is heavily accumulated in the soma, nucleus and nucleolus of *App^NL-F^* neurons cultured for 7 and 14 DIV, while only the nucleolus showed an accumulation at 21 DIV. These data indicate that the increased levels of DDX24 occur early during the culturing of hippocampal neurons but disappear when the neuron is mature, which reflects the change of DDX24 in the brains of *App^NL-F^* mice. Furthermore, DEAD Box Helicase 17, a similar protein to DDX24, has been implicated in amyloidosis, suggesting that the DEAD box helicase family might influence AD pathology [25].

While 6-month-old *App^NL-F^* mice, which have excessive amounts of Aβ42 due to increased production, had increased intraneuronal DDX24 levels, high levels of Aβ42 added to the cell culture medium of mouse primary hippocampal neurons resulted in decreased intraneuronal DDX24 levels as well as decreased APP levels. Interestingly, the diminished effect of DDX24 accumulation with age in the *App^NL-F^* mouse brain was mimicked by the diminished effect of DDX24 accumulation in cultured neurons at 21 DIV, indicating that too high levels of Aβ42 have the opposite effect. There is also a possibility that the effect of Aβ42 on DDX24 levels is different for endogenous as compared to endocytosed Aβ42. In support of this hypothesis, our previous findings have shown that Aβ42 taken up into neurons by endocytosis over time accumulates in late endosomes/lysosomes in soma, where it aggregates, while endogenously produced Aβ42 in neurons is present in other subcellular locations, such as the presynaptic side of the synapse [26]. Thus, Aβ42, which is believed to have physiological effects on synaptic functions at low concentrations [27], whereas it becomes toxic and aggregates when concentrations are too high [28], appears to have different effects on neurons under different conditions. It is of note that Aβ40 levels in CSF of the subtype of AD patients characterised by RNA dysregulation did not differ significantly from control [20].

Interestingly, the decreased levels of APP and Aβ that we observed as a result of DDX24 siRNA silencing in cortical neurons suggest that there might be a mechanistic link between DDX24 and APP expression and processing. DDX24 silencing reduced the number of synapses, indicating a possible role of DDX24 in synaptogenesis and axonal maintenance. It is possible that DDX24 interacts with APP metabolism to influence the production of Aβ but is negatively regulated by Aβ itself. There are several other potential ways that DDX24 could affect neurons in AD through mediation with other potential AD regulators. For example, dysregulation of p53 is associated with ageing and has been suggested as a mediator in AD pathology [16,29]. DDX24 has been shown to increase p53 function via Mouse double minute 2 homolog, implicating a potential mechanistic relationship [14,30,31]. Lastly, since the DEAD-box helicase family is also heavily involved in mRNA processing and highly conserved, a speculative argument could be made that ribosomal dysfunction in AD might be influenced by DDX24 expression [32,33]. Our findings in a mouse model for AD showed increased DDX24 levels before the onset of the disease, a change that did not persist throughout ageing. This could be indicative of a regulatory effect on the disease pathology early during the disease progression. Based on our data, we propose the existence of a reciprocal relationship between DDX24 and APP metabolism and Aβ42 generation and thus AD pathology, which emphasises the importance of elucidating its role in the progression of the disease.

In conclusion, by using a combination of proteomics, pathway analysis, IHC, confocal microscopy and cell culture studies, DDX24 was identified as an influencer on AD pathology in both mice and humans. DDX24 levels were increased in AD brain and in presymptomatic *App^NL-F^* mice. Silencing of DDX24 reduced APP and Aβ42 levels, while the addition of Aβ42 to the medium of primary neurons resulted in lowered DDX24 levels. Thus, there are implications of a mechanistic relationship between APP and DDX24, and future studies should focus on evaluating this mechanism.

## 4. Materials and Methods

### 4.1. Preparation of Postmortem Brain Homogenates

Frozen dissected tissue of the hippocampus and frontal cortex from five AD and five control postmortem brains were obtained from the Netherlands Brain Bank (NBB), Netherlands Institute for Neuroscience, Amsterdam. The samples were homogenised in 8 M urea (Sigma-Aldrich, Saint Louis, MO, USA) and 500 mM ammonium bicarbonate (Fluka 40867) at a final concentration of 100 mg/mL, as described previously [34]. The procedures for handling the postmortem human samples were in accordance with the ethical standards of the institutional and national regional research committee “Regionala etikprövningsnämnden i Stockholm” (ethic permit nr 2013/1301-31/2) and with the 1964 Helsinki declaration and its later amendments or comparable ethical standards. The samples were sex- and age-matched, and the control subjects had no known psychiatric or neurological disorders (Table 2).

### 4.2. Proteomics

Hippocampal or cortical homogenates in aliquots corresponding to 50 μg protein (determined by the BCA protein assay, Pierce) from each of the control and AD cases were digested by trypsin as described previously [14]. Internal standards for the hippocampal and cortical samples were prepared by tryptic digestion of a mixture of all the samples from each brain region in the presence of 97% ^18^O-labeled H_2_O. Each sample was mixed with the internal standard at a 1:1 ratio, fractionated into four fractions by using ion-exchange pipette tips and subjected to LC-MS/MS analysis on a Q Exactive instrument (ThermoScientific, Waltham, MA, USA) [14].

### 4.3. Ingenuity Pathway Analysis

*T*-tests were performed with two tails and two samples with unequal variance, and the resulting file containing a protein identification number, fold change in protein levels, and *p*-values were uploaded to IPA. The following settings were used for analysis: Reference set: in ingenuity knowledge base. Consider only molecules and/or relationships where (species = Rat OR Mouse OR Human) AND (confidence = Experimentally Observed) AND (tissues/cell lines = Cerebral Cortex OR Melanocytes OR Vascular smooth muscle cells OR Striatum OR SF-295 OR Nervous System not otherwise specified OR Caudate Nucleus OR SNB-75 OR Pituitary Gland OR SF-539 OR Purkinje cells OR Brainstem OR Other Epithelial cells OR Microvascular endothelial cells OR Amygdala OR Ventricular Zone OR Subventricular Zone OR Spinal Cord OR Blood platelets OR Olfactory Bulb OR Sertoli cells OR Astrocytes OR Cerebral Ventricles OR Cortical neurons OR Corpus Callosum OR Dorsal Root Ganglion OR White Matter OR Sciatic Nerve OR Neurons not otherwise specified OR Putamen OR Thalamus OR SF-268 OR Trigeminal Ganglion OR Other Nervous System OR Keratinocytes OR Hepatocytes OR Endothelial cells not otherwise specified OR Nucleus Accumbens OR CNS Cell Lines not otherwise specified OR Other CNS Cell Lines OR Other Endothelial cells OR Brain OR Granule cells OR Choroid Plexus OR Granule Cell Layer OR Medulla Oblongata OR Epithelial cells not otherwise specified OR Microglia OR Hypothalamus OR Hippocampus OR U87MG OR Cerebellum OR Pyramidal neurons OR Parietal Lobe OR HUVEC cells OR Substantia Nigra OR Gray Matter OR Other Neurons OR U251 OR Other Tissues and Primary Cells) AND (data sources = An Open Access Database of Genome-wide Association Results OR BIND OR BioGRID OR Catalogue Of Somatic Mutations In Cancer (COSMIC) OR Chemical Carcinogenesis Research Information System (CCRIS) OR ClinicalTrials.gov OR ClinVar OR Cognia OR DIP OR DrugBank OR Gene Ontology (GO) OR GVK Biosciences OR Hazardous Substances Data Bank (HSDB) OR HumanCyc OR Ingenuity Expert Findings OR Ingenuity ExpertAssist Findings OR IntAct OR Interactome studies OR MIPS OR inholee OR miRecords OR Mouse Genome Database (MGD) OR Obesity Gene Map Database OR Online Mendelian Inheritance in Man (OMIM) OR TarBase OR TargetScan Human). Set cutoffs: Experimental *p*-value of *p* < 0.1 and experimental fold change 1.1 times up- or downregulated.

### 4.4. Mouse Model

The *App^NL-F^* mouse model is a single APP knock-in mouse, which harbours both the Swedish and Beyreuther/Iberian APP mutations [22]. This mouse model produces typical Aβ associated pathology and symptoms. WT and *App^NL-F^* mice, both on C57BL/6 background, were used for immunohistochemistry and for the preparation of primary neuron cultures. The mice were housed in 12 h dark light transition with ad libitum access to food. All *App^NL-F^* mice used were homozygous for the *App^NL-F^* knock-in (*App^NL-F/NL-F^*).

### 4.5. Brain Tissue for Immunohistochemistry

#### 4.5.1. Human

Human brain samples containing the hippocampus and frontal cortex from AD (*n* = 15) and control patients (*n* = 15) were obtained from the Netherlands Brain Bank (NBB). All AD subjects met the criteria for definitive AD according to the Consortium to Establish a Registry for AD [35]. The control subjects had no known psychiatric or neurological disorders. Table 1 displays detailed information about the gender distribution, age at death, neuropathological diagnosis, brain weight, post-mortem processing intervals (PMI), Braak staging, and brain bank origin of the donors (Table 3). Samples were paraffinised in whole tissue sections, cut and mounted on glass slides.

#### 4.5.2. Mouse

Mouse brain were collected from the *App^NL-F^* (*n* = 5) and wild type (WT) (*n* = 5) mice, both on C57BL/6 background (Ethical approval from Stockholms djurförsöksetiska nämnd: Dnr 16334-2022). The brains were mounted in paraffin on glass slides. To get information about age-dependent changes in protein expression, mice brains were examined at 6 months, 12 months, and 25 months. Mouse brains were stained in the same way as human samples, as described below.

### 4.6. Primary Neuronal Culturing

Mouse embryos (16–17 days) were taken from pregnant WT and *App^NL-F^* mice (Ethical approval from Stockholms djurförsöksetiska nämnd: Dnr 16334-2022) and stored in cold Hank’s balanced salt solution buffer. The cortex and hippocampus were dissected as previously described [27]. Cortical or hippocampal neurons were seeded at the centre of a 35 mm glass bottom culture dish (P35-G-1.5-10.C; MatTek, Boston, MA, USA) pre-coated with poly-D-lysine. A support layer of cortical neurons was seeded at the edges of the plate. Neurobasal medium with 2% B27 and 1% L-glutamine was used for culturing. Neurons were fixed after 7, 14 and 21 DIV and stained with antibodies for confocal microscopy. Each batch of neuronal cultures included two to three embryo brains without distinction between male and female mice (*n* = 3–5).

### 4.7. Immunohistochemistry

A thin section (8 µm) of paraffinised brain tissue samples containing hippocampus from humans and *App^NL-F^* or WT mice were deparaffinised and hydrated, first in xylene and then in a decreasing concentration of ethanol in water using 99.5%, 95%, and 70% ethanol *w*/*v* before placing in distilled water. Samples were autoclaved for antigen retrieval in a DIVA decloaked bath at 110 °C for 20 min before being washed with dH_2_O. Subsequently, the protocol for the IHC kit Envision+ MACH1 universal labelling system (BioCare Medical, Pacheco, CA, USA) was adapted from the manufacturer’s protocol. The background of the samples was blocked by peroxidase treatment at room temperature (RT) for 10 min, followed by washing with phosphate-buffered saline containing 0.05% Tween20 (PBS-T), three times for five min each with gentle rocking. Samples were then incubated overnight (ON) at 4 °C with primary anti-DDX24 antibody, HPA 002554, 1:100, (Sigma-Aldrich, Saint Louis, MO, USA) in PBS + 4% NGS. This antibody was validated by siRNA silencing in primary neuronal cultures, followed by immunocytochemistry quantification. For negative control samples, the primary antibody was omitted. After washing the slides three times for five min in PBS-T, secondary labelling was performed using HRP-conjugated polymer antibody for 2 h (h) at RT. After this secondary incubation step, samples were washed with PBS-T three times for five minutes before being incubated with a DAB-Chromogen solution for five min at RT. Samples were rinsed with distilled water before being dehydrated in an increasing concentration of ethanol and put in a xylene bath before mounting. Samples were mounted using a xylene-based mounting medium Vectamount (Thermofisher, Waltham, MA, USA), covered with cover slip panels with a refractive index of one and observed using light microscopy within 48 h of mounting. Samples were stored dry and dark when not in use.

### 4.8. Immunofluorescence Microscopy

The cellular localisation of DDX24 was studied using confocal microscopy. Paraffinised human brain tissue slides were deparaffinised and treated for antigen retrieval as described above. Samples were washed three times for five min in PBS-T and incubated with primary antibodies, anti-DDX24 (1:100) HPA 002554 with the immunogen ATP-dependent RNA helicase DDX24 recombinant protein epitope signature tag (PrEST) (Sigma-Aldrich, Saint Louis, MO, USA), anti-APP C-terminal C1/6.1 which have the epitope within amino acids 45–53 of APP (1:200), C1/6.1(BioLegend, San Diego, CA, USA), anti-amyloid precursor protein Y188 ab32136 (Abcam, Cambridge, UK) in PBS overnight (ON) at 4 °C. For negative control samples, the primary antibody was omitted. After washing three times for five min with PBS-T, a secondary incubation step was performed at RT 1 h with directly labelled secondary antibodies Alexa flour 647 and Alexa flour 555 at 1:500 in PBS (Sigma Aldrich, Saint Louis, MO, USA). After washing three times for five min, mounting was performed using ProLong gold anti-fade reagent P36930 (Life Technologies, Carlsbad, CA, USA), and slides were covered with coverslips with number 1.5. Samples were stored in darkness and at 4 °C when not in use.

#### 4.8.1. DDX24 Quantification and Localisation in Neurons

Neuronal cultures were fixed using 4% formaldehyde (Sigma-Aldrich, Saint Louis MO, USA), HT5011, for 10 min at RT before being washed with 1 mL PBS three times. Cells were then permeabilised with 0.4% CHAPSO in PBS for 10 min at RT. Cultures were blocked with normal goat serum for 15 min at RT. Primary antibody incubation with anti-DDX24 (1:100) HPA 002554 (Sigma-Aldrich, Saint Louis, MO, USA) and anti-nuclear pore complex (1:1000), FG-Nups (BioLegend, San Diego, CA, USA) was performed ON at 4 °C. After incubation, plates were washed with PBS, three x five min, before being incubated with secondary antibodies Alexa Fluor 647 (1:500), Alexa Fluor 555 (1:500), and phalloidin (1:200) in 3% NGS in PBS for 2 h at RT. The cultures were washed with PBS-0.1% Tween three x five min and then with PBS before being postfixed with 3% formalin and 0.1% glutaraldehyde in PBS for 10 min at RT. Samples were washed with PBS, three x five min, washed with ddH_2_O and then mounted with Vectashield antifade mounting medium (Vector Laboratories, Burlingame, CA, USA) and stored in darkness until imaging.

#### 4.8.2. Co-Staining with APP

The fixation and staining of neuronal cultures were performed as described above. Primary antibody incubation was conducted with anti-DDX24 (1:100) HPA 002554 (Sigma-Aldrich, Saint Louis, MO, USA) and anti-APP C-terminal (1:200), C1/6.1 (BioLegend, San Diego, CA, USA) ON at 4 °C. Secondary antibodies used were Alexa Fluor 555 (1:1000), Alexa Fluor 647 (1:1000), and phalloidin (1:200) in 3% NGS in PBS.

### 4.9. Image Acquisition

Fluorescently stained IHC images were acquired in sequential mode setting on a Nikon A1Rsi point laser scanning confocal inverted microscope (Nikon, Tokyo, Japan) using 20× or 60× objectives and an image size of 1024 × 1024 pixels. Image capturing was performed via the attached image acquisition software, Nikon NIS Elements (https://nivina.com.vn/vn/images/media/goc/1461216578NIS%20Elements%202012.09.pdf, accessed on 5 July 2023) (Nikon, Tokyo, Japan). Excitation lasers were 561 and 640 nm. The settings, including laser power intensities, pinhole and detector gain, were chosen to optimise the dynamic range to show no fluorescence signal in the negative control and limited or no saturation in the strongest signal localisations. The pinhole size was set to one airy unit for the far red and green channels.

IHC pictures of DAB-chromogen-stained brain sections were captured using a Nikon Camera DS-Qi2 (Nikon, Tokyo, Japan) with capture software Nikon NIS-elements. Colour correction channels were adjusted for whitening, and conditions were kept the same for each image capture. CA1, CA3 and the frontal temporal cortex were imaged and analysed. Images were captured using 10×, 20×, and 40× objectives.

Confocal images of neuronal cultures were captured using a Zeiss LSM 900 or Zeiss LSM 980 with Airyscan 2 using the ZEN version 3.6 software (Zeiss, Stockholm, Sweden). Images for siRNA-treated neurons were captured using 20× and 40× objectives with an image size of 1024 × 1024 pixels. Images of neuronal cultures used for DDX24 quantification and localisation were taken using a 63× objective with pixel size set to optimal. Excitation lasers with 488, 561 and 640 nm were used. The settings, including laser power intensities, pinhole and detector gain, were chosen to optimise the dynamic range to show no fluorescence signal in the negative control and limited or no saturation in the strongest signal localisations. The pinhole size was set to one airy unit.

### 4.10. siRNA Treatment of Primary Neuronal Cultures

At 11 days in vitro, neuronal cultures were incubated with siRNA against DDX24 (ThermoFischer, Waltham, MA, USA) using the Lipofectamine RNAiMAX protocol (ThermoFischer, Waltham, MA, USA). Five pmol of siRNA were used per well, and cultures were allowed to incubate for 24 h before fixation and staining (ThermoFischer, Waltham, MA, USA).

### 4.11. Lysate of Primary Neuronal Cultures

Primary neuronal cultures intended for ELISA analysis were subjected to lysis using RIPA buffer. The medium was carefully removed from the adherent cells, and the cells were washed with PBS. PBS was removed, and chilled RIPA lysis buffer was added to the cells. The cells were incubated on ice for 30 min before being collected and centrifuged at 14,000× *g* for 10 min. The supernatant was resuspended in PBS prior to ELISA analysis.

### 4.12. Treatment of Primary Neuronal Cultures with Aβ42

1 μmol/mL of Aβ42 was added to the medium of 11 DIV primary neuronal cultures and allowed to incubate for 24 h. Primary neuronal cultures were then fixed and imaged as described above.

### 4.13. ELISA

ELISA was used for quantitative determination of Aβ in both neuronal culture medium and lysate of primary neurons. Medium and neuronal cell lysate from siRNA-treated primary cultures were subjected to ELISA analysis using the β-amyloid 42 ELISA kit according to the manufacturer’s assay protocol, Code no. 292-64501 (Wako, Richmond, VA, USA). The ELISA kit uses the monoclonal antibody BNT77, which targets Aβ 11–28 and the monoclonal antibody BC05, which targets the C-terminal portion of Aβ42. ELISA plates were read using a CLARIOStar^Plus^ plate reader (BMGLABTECH, Ortenburg, Germany). All ELISA experiments were repeated to ensure reproducibility, *n* = 3.

### 4.14. Quantification and Statistics

Quantification of signal intensity in IHC images was performed using ImageJ imaging software, version 1.54c (NIH, Lyford, UK). A positive signal was detected based on the mean signal intensity in each sample area. Regional signal was defined as the whole signal detectable in the anatomically identified area. The neuronal signal was defined as the average signal of 40 neurons per subarea. The signal intensity was displayed in an interval of 0–250. The signal in neuronal cultures was measured by drawing a mask around the area of interest and calculating the mean signal intensity. The phalloidin signal was used as a morphological marker for the calculation of the number of spines.

Statistical analysis of signal intensity data obtained via ImageJ was performed using Graphpad Prism software (Graphpad Software, version 24.0, La Jolla, CA, USA). Unpaired Students *t*-tests were used to determine the significance between intensity signals with a *p*-value < 0.05 considered significant. In case of a violation of the normality assumption, the Mann-Whitney U test was used instead.

## Figures and Tables

**Figure 1 ijms-25-03622-f001:**
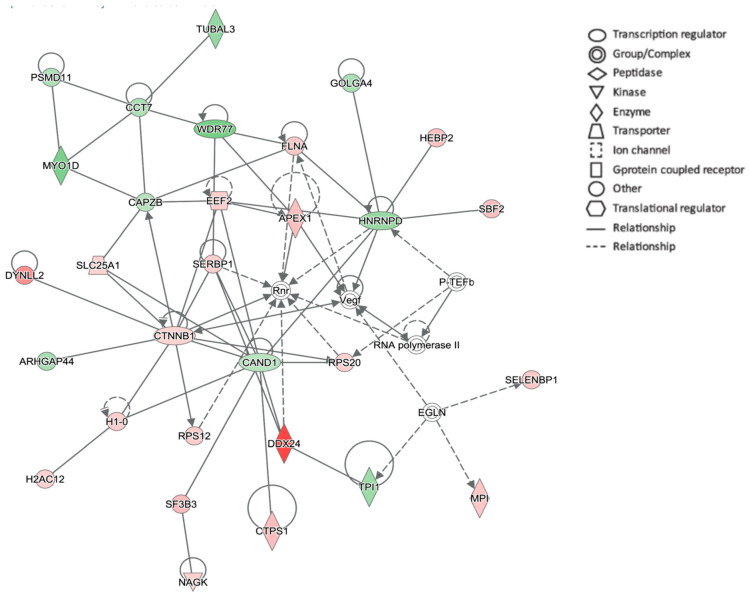
The network identified with the highest score included DDX24 as a player in AD pathology. The network is called “Cellular assembly and organisation, cancer, developmental disorder”. Red, upregulated in AD; Green, downregulated in AD. A higher intensity indicates a higher fold change in AD compared to ND. Arrows indicate relationship. A straight line indicates a direct relationship. A dotted line indicates an indirect relationship. More information on the figure legend can be found at https://qiagen.my.salesforce-sites.com/KnowledgeBase/articles/Knowledge/Legend (accessed on 18 March 2024).

**Figure 2 ijms-25-03622-f002:**
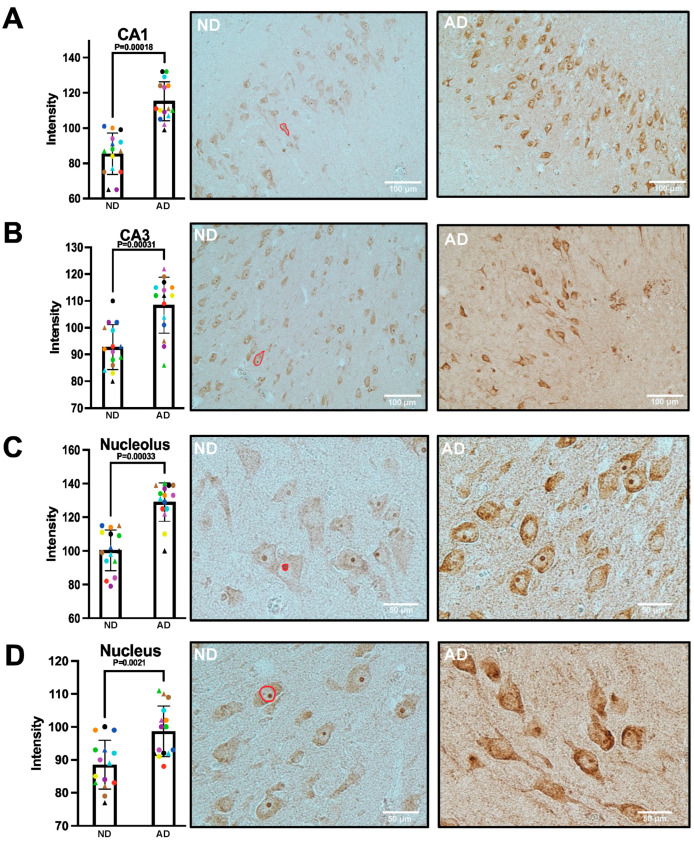
DDX24 immunohistochemistry and quantification in human hippocampus. Thin sections from AD and ND brains were stained with antibodies directed to DDX24 as described in Materials and Methods. DDX24 is shown in neurons in (**A**) CA1 and (**B**) CA3 regions of the hippocampus. DDX24 accumulations could be found both in (**C**) the nucleolus and (**D**) the nucleus of pyramidal neuronal cells. Red lines indicate examples of analysed areas. The different colours represent unique individuals. Quantification was made using Image J software version 1.54d.

**Figure 3 ijms-25-03622-f003:**
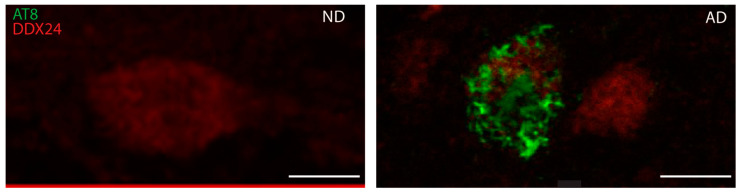
Co-staining of DDX24 and phosphorylated tau in AD and ND brain. Confocal microscopy of thin sections from the hippocampus in AD and ND brain were stained with an antibody directed to DDX24 and AT8 antibody. The latter only binds to phosphorylated tau. Red, DDX24; Green, AT8. The scale bar is 20 µm.

**Figure 4 ijms-25-03622-f004:**
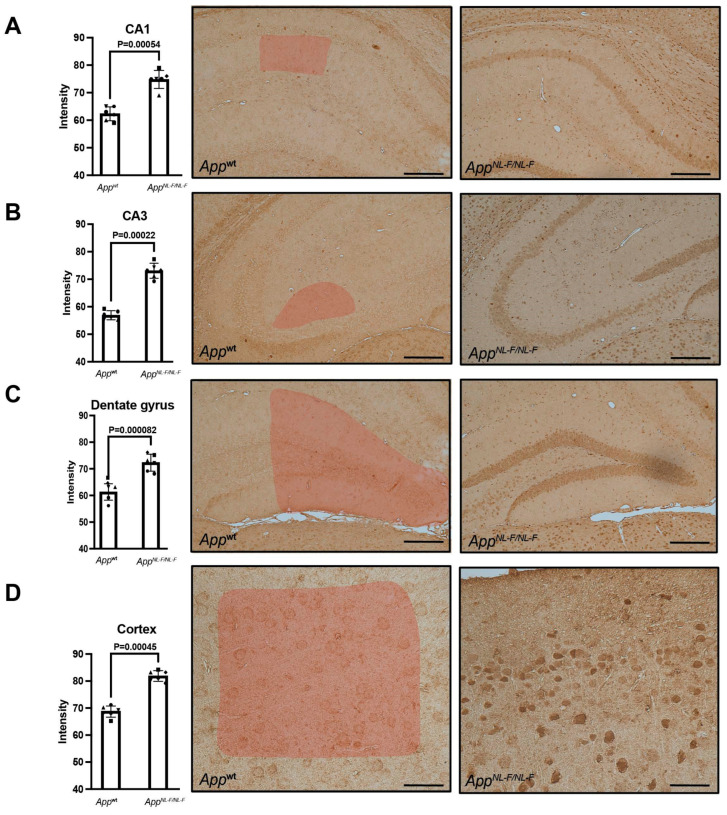
DDX24 immunohistochemistry and quantification in *App^NL-F^* and wild-type mouse brain. Thin sections from 6-month-old *App^NL-F^* and wild-type (WT) mice were stained with antibodies directed to DDX24 as described in materials and methods. High levels of DDX24 were detectable in neurons in the (**A**) CA1 and (**B**) CA3 regions of the hippocampus. (**C**) The granule cell layer of the dentate gyrus. (**D**) The cortex of *App^NL-F^* mice also contained DDX24 in the neuronal cells. Quantification was made by using Image J software. Bar graph symbols indicate individual mice. The read area indicate area analyzed. The scale bar is 50 µm (**A**–**C**) and 25 µm (**D**).

**Figure 5 ijms-25-03622-f005:**
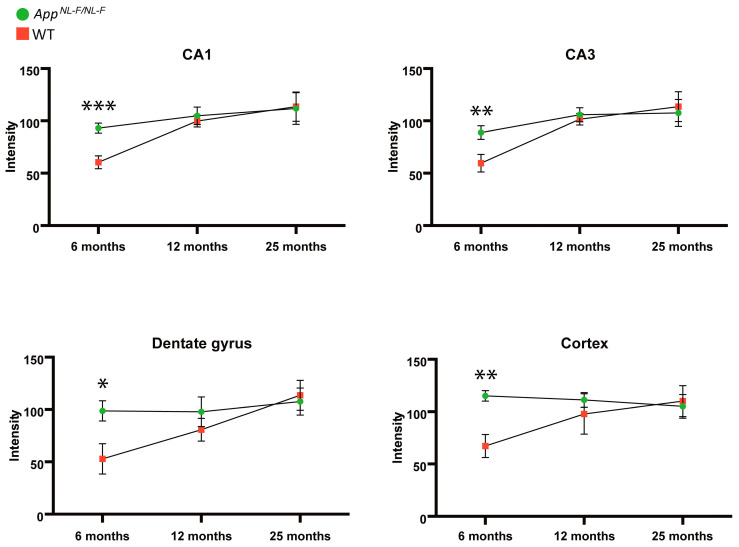
Effects of age on DDX24 levels in the brain from *App^NL-F^* mice compared to wild-type mice. Immunohistochemistry was performed on thin sections from 6-, 12-, and 25-month-old *App^NL-F^* and wild-type (WT) mice, using an antibody directed to DDX24. Significant differences between *App^NL-F^* and WT at a specific time point are indicated in the graphs. Quantification of the indicated regions was made using Image J software, and the relative levels were plotted for hippocampal regions CA1, CA3, and dentate gyrus, as well as for the cortex. Images are shown in Appendix A. * = *p* < 0.05, ** = *p* < 0.01, *** = *p* < 0.001.

**Figure 6 ijms-25-03622-f006:**
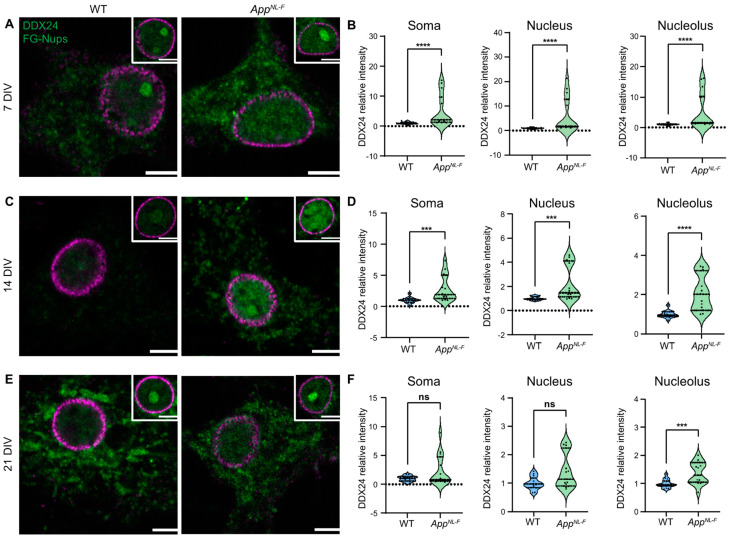
Confocal microscopy of DDX24 in primary neuronal cultures derived from *App^NL-F^* and WT mice. Primary neurons were cultured for 7 (**A**,**B**), 14 (**C**,**D**), and 21 (**E**,**F**) DIV, and the intensity of DDX24 was measured via confocal microscopy in the soma, nucleus, and nucleolus. DDX24 and nuclear marker are shown in green and magenta, respectively. Scale bars are 5 µm. ns = *p* > 0.05, *** = *p* < 0.001, **** = *p* < 0.0001.

**Figure 7 ijms-25-03622-f007:**
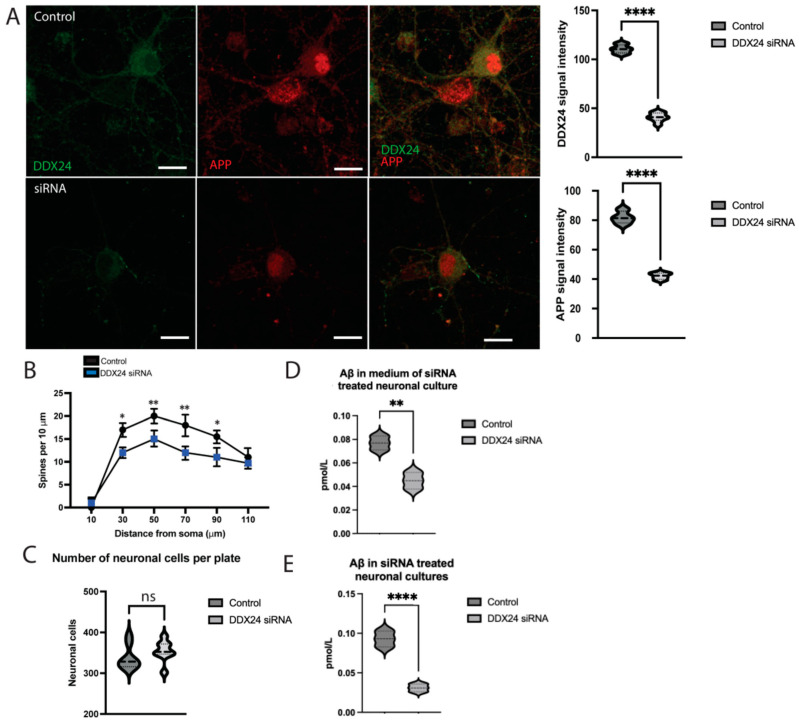
Effects of siRNA silencing of DDX24 on APP and Aβ42 levels and spine density in primary neurons. Wild-type (WT) mouse cortical neurons were treated with siRNA targeting DDX24, followed by immunofluorescence and confocal microscopy or Aβ42 ELISA of the cell culture medium and cell lysates. (**A**) Signal intensity of APP and DDX24 was compared between cortical neurons with and without siRNA treatment. We also compared (**B**) spine density, (**C**) neuronal count per plate and Aβ42 in both (**D**) medium and (**E**) neuronal cell lysate. DDX24 is shown as green and APP as red in confocal microscopy images. The scale bar is 50 µm. ns = *p* > 0.05, * = *p* < 0.05, ** = *p* < 0.01, **** = *p* < 0.0001.

**Figure 8 ijms-25-03622-f008:**
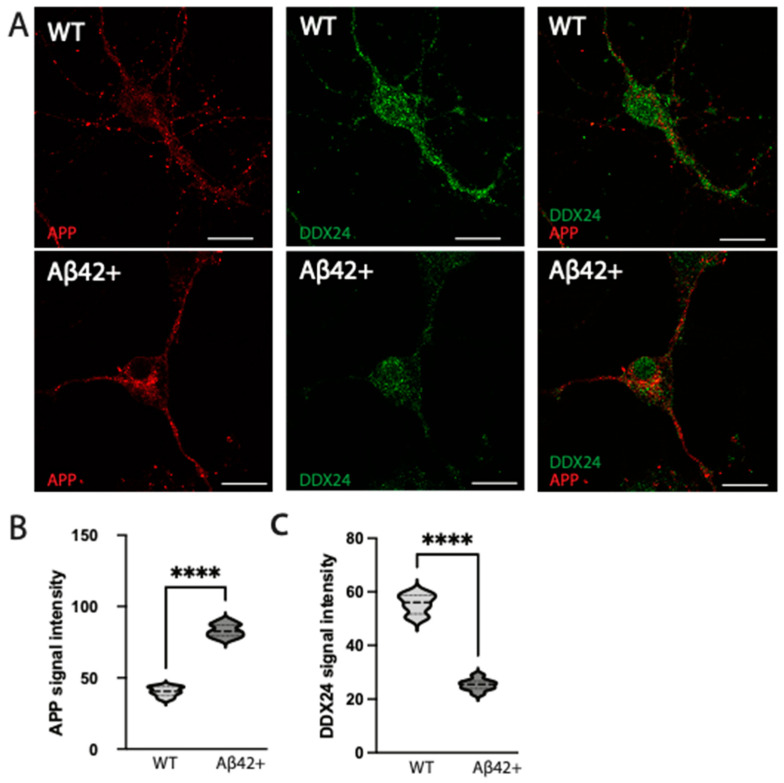
Confocal microscopy of primary neuronal cultures exposed to a high concentration of Aβ42. Primary neuronal cultures were incubated for 24 h with 1 mmol/mL of Aβ42 (**A**). DDX24 and APP signal intensity were measured and compared between Aβ42 positive neuronal cultures (Aβ42+) and control without extra Aβ42 (Aβ42−) (**B**,**C**). **** = *p* < 0.0001. The scale bar is 50 μm.

**Table 1 ijms-25-03622-t001:** Proteins with highest change in levels in AD compared to control brain. Upwards-pointing arrows indicate an increase in protein levels, and downwards-pointing arrows indicate a decrease in protein levels.

Hippocampus (AD/Cntr)			Cortex (AD/Cntr)		
Uniprot Code	Molecule	Fold Change	Uniprot Code	Molecule	Fold Change
Q9GZR7	ATP-dependent RNA helicase DDX24	↑ 11.0	Q9Y316	Protein MEMO1	↑ 6.43
P49840	Glycogen synthase kinase-3 alpha	↑ 4.57	Q16348	Solute carrier family 15 member 2	↑ 3.48
Q96RT7	Gamma-tubulin complex component 6	↑ 3.78	Q9P2K2	Thioredoxin domain-containing protein 16	↑ 3.00
Q96FJ2	Dynein light chain 2, cytoplasmic	↑ 3.45	P24588	A-kinase anchor protein 5	↑ 2.47
P04233	HLA class II histocompatibility antigen gamma chain	↑ 3.25	P55957	BH3-interacting domain death agonist	↑ 2.35
P0C646	Olfactory receptor 52Z1	↑ 2.99	P02786	Transferrin receptor protein 1	↑ 2.34
P41222	Prostaglandin-H2 D-isomerase	↑ 2.96	P63098	Calcineurin subunit B type 1	↑ 2.29
Q8N6D5	Ankyrin repeat domain-containing protein 29	↑ 2.75	Q8NE71	ATP-binding cassette sub-family F member 1	↑ 2.27
Q9HCP6	Protein-cysteine N-palmitoyl-transferase HHAT-like protein	↑ 2.69	Q9C0C2	182 kDa tankyrase-1-binding protein	↑ 2.22
Q9UBS4	DNAJB11 DnaJ homolog sub-family B member 11	↑ 2.58	Q9H444	Charged multivesicular body protein 4b	↑ 2.06
P23515	Oligodendrocyte-myelin glycoprotein	↓ −4.26	Q9P0S9	Transmembrane protein 14C	↓ −3.70
P02686	Myelin basic protein	↓ −4.17	P26639	Threonine—tRNA ligase, cytoplasmic	↓ −2.94
P60201	Myelin proteolipid protein	↓ −3.59	O75508	Claudin-11	↓ −2.94
O75508	Claudin-11	↓ −2.85	Q9H228	Sphingosine 1-phosphate receptor 5	↓ −2.86
Q9BQA1	Methylosome protein 50	↓ −2.60	P34897	Serine hydroxy-methyltransferase, mitochondrial	↓ −2.44
P59768	Guanine nucleotide-binding protein G(I)/G(S)/G(O) subunit gamma-2	↓ −2.48	P24534	Elongation factor 1-beta	↓ −2.38
O94832	Unconventional myosin-Id	↓ −2.39	P02511	Alpha-crystallin B chain	↓ −2.22
Q9H0E2	Toll-interacting protein	↓ −2.30	Q9H902	Receptor expression-enhancing protein 1	↓ −2.17
Q14982	Opioid-binding protein/cell adhesion molecule	↓ −2.22	P21359	Neurofibromin	↓ −2.17
O75363	Breast carcinoma-amplified sequence 1	↓ −2.21	Q15386	Ubiquitin-protein ligase E3C	↓ −2.08

**Table 2 ijms-25-03622-t002:** Patient information for AD and control samples used for proteomics.

Variable		Non-Demented Controls (*n* = 5)	AD (*n* = 5)
Age of death		79 ± 6	81 ± 5
Number of females		3	3
PMD (hours)		6.8 ± 2.0	6.3 ± 1.1
Braak scores	0	1	0
	I/II	4	0
	III/IV	0	1
	V/VI	0	4

**Table 3 ijms-25-03622-t003:** Patient information for AD and control samples used for immunohistochemistry.

Clinical Diagnosis	Braak	AD (*n* = 15)	Control (*n* = 15)
Age of death (years), mean, SD (range)		82 ± 2.2 (72–89)	83 ± 6.3 (76–93)
Number of females		8	8
Brain weight (g), mean, SD (range)		985 ± 100.8 (843–1135)	1119 ± 122 (1001–1361)
Post-mortem interval (hours), mean, SD (range)		4.3 ± 0.7 (3–5)	5.5 ± 1.3 (5–7.4)
Distribution of Braak scores	0	0	11
	I/II	0	4
	III/IV	0	0
	V/VI	15	0
Brain bank	NBB KBB	9 6	9 6

## Data Availability

All data generated or analysed during this study are included in this published article and its Appendix A.

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
