# Peer review of "DEAD Box Helicase 24 Is Increased in the Brain in Alzheimer’s Disease and AppN-LF Mice and Influences Presymptomatic Pathology"

_ijms, 2024, doi:10.3390/ijms25073622_

Round 1

Reviewer 1 Report

Comments and Suggestions for Authors

In the present research work, authors tried to indicate the role of DDX24 in the pathology of AD in initial stages. Here are some of my observations.

- The title of the manuscript may be modified as the term,Älzheimer disease brain'in title does not seem suitable. The authors may modifiy it for better understanding of readers.

- The authors should provide the reference for line no 65-68.

- In fig 2 (a-d), What authors want to indicate with different color symbol in bar diagrams. It should be clarified in legend or caption of figure.

- Why authors did not specify the analysed area in AD samples (fig 2)

- In fig 4, Analysed area should be specified. 

- DDX24 level increase in brain samples of App (NL-F) mice of 6 months as compared to older age animals, whereas is level is almost similar to WT animals of 12 or 25 months age. If DDX24 is highly regulating the Amyloid Beta production then why it is not affecting in old age animals. 

- As per the discussion, the authors  justified that DDX24 is involved in initial phases of immune response and in mice Aβ42 accumulated from birth and that is why it is connected with AD detection in early stages in mice only. However, if that is the case, then how could it be related to early diagnosis of AD in humans if  Aβ42 starts accumulating in humans in the late stages of life? 

Comments on the Quality of English Language

Minor changes are required. 

Reviewer 2 Report

Comments and Suggestions for Authors

This is certainly an interesting paper with an interesting finding.  The approach is not innovative but the list of proteins discovered seems unique.  The paper tries to cover all angles of this discovery of the protein DDX24 but does leave out some of the fundamentals.  I pose the following questions that l view as issues that the authors might like to discuss or provide additional data.

I should add that I am not able to access the supplementary figures that are mentioned in the text.  These are not available to download on the website l am using.!!

1.   I would suggest a greater discussion on why the authors think DDX24 might have a critical role in AD pathology.  This is only covered superficially in the discussion, which is mainly a repetition of the findings.  Could the authors provide detailed discussion of whether other papers (whether proteomics or genomics) have identified this protein, or whether it was not found to be different in others.

2.  As this protein was only featured on the hippocampus samples and not the cortex samples, this requires some discussion.  I would have some concern about this as it relates to AD pathology.  It is well established that many ND elderly have extensive tangle pathology in the HPC, even if not much amyloid, but the neuropathology diagnosis is often dependent on the amyloid load in the cortical regions.  Although the authors have provided some details, it would be interesting to know the degree of tangle pathology in the hippocampus of the ND samples specifically.  Do you have any phosphotau images of these samples taken before initial experiments.  Related to that, the protein list between HPC and Cortex is almost completely different.  This deserves a comment.

3.  I have two concerns concerning the immunohistochemistry.  Firstly, there is no validation of the DDX24 antibody presented.  How specific is it under the conditions used for IHC.  As a minimum, a western blot is needed to show its specificity.  My own experiences with the sigma HPA antibodies do not always match the specificity claims presented by the manufacturers.  Secondly, a concern to me is the difference in background staining between the ND samples and the AD samples.  There are dramatically different levels of DAB staining between the ND and AD samples.  I would view this as background staining as much of it is in neuropil and not cell bodies.  To convincingly demonstrate this difference is real and not technical, some western blot validation of ND and AD samples are needed.  This can be quantitative or semi-quantitative.  I can not accept that this is proven by the proteomics data alone without validation.  Fig 4 D also has differential background, while all of the figures in Fig 4 have very high background.

4.  The control for the IHC is not adequate.  Deleting a primary antibody is not a valid control.  Need to substitute another irrelevant antibody (rabbit IgG?) at the same concentration as used in the DDX24 stained sections.  This is particularly important as my experience has been that nucleolus staining is often artefact.  Some adequate controls needed.  I do not accept that the ImageJ software adequately accounts for differential background.

5.  The bar charts and figures in Figure 2 should be in the same orientation. Right now the AD on left in bar charts and on right in Figures.

6.  I do not think it appropriate to use the term "control" or "Cntr" for elderly humans.  These should be described simply as ND - non demented.  These samples are from elderly subjects who have passed away so they are not controls for anything.

7.  The authors data on the difference due to the effect of aging in Tg mice for DDX24 expression disappears at 12 months.  This deserves some discussion.

4.

Round 2

Reviewer 1 Report

Comments and Suggestions for Authors

In the discussion of results , the authors should justify the results with cited references.

Author Response

Reviewer #1 

In the discussion of results , the authors should justify the results with cited references. 

Response 

We have clarified in the discussion when we are talking about our results (p. 16, line 308-310), and have added some additional references when we are discussing, for instance, the AppNL-F mouse model.  

Reviewer 2 Report

Comments and Suggestions for Authors

The authors have decided not to include the essential controls requested.  The responses do not really improve the paper.

Author Response

Reviewer #2 

The authors have decided not to include the essential controls requested. The responses do not really improve the paper. 

Response 

We have included some additional information in the manuscript on how we validated the antibody using siRNA silencing as well as how we believe the background staining is mostly staining in neurites. Even though we did not include some of the experiments the reviewer requested, we hope to have addressed the comments more clearly and transparently in the manuscript.